# Learnable Graph Convolutional Attention Networks

## Abstract

Existing Graph Neural Networks (GNNs) compute the message exchange between nodes by either aggregating uniformly (*convolving*) the features of all the neighboring nodes, or by applying a non-uniform score (*attending*) to the features. Recent works have shown the strengths and weaknesses of the resulting GNN architectures, respectively, GCNs and GATs. In this work, we aim at exploiting the strengths of both approaches to their full extent. To that end, we first introduce a graph convolutional attention layer (CAT), which relies on convolutions to compute the attention scores. Unfortunately, as in the case of GCNs and GATs, we then show that there exists no clear winner between the three—neither theoretically nor in practice—since their performance directly depends on the nature of the data (i.e., of the graph and features). This result brings us to the main contribution of this work, the learnable graph convolutional attention network (L-CAT): a GNN architecture that allows us to automatically interpolate between GCN, GAT and CAT in each layer, by only introducing two additional (scalar) parameters. Our results demonstrate that L-CAT is able to efficiently combine different GNN layers across the network, outperforming competing methods in a wide range of datasets, and resulting in a more robust model that needs less cross-validation.

## 1  Introduction

In recent years, Graph Neural Networks (GNNs) [25] have become ubiquitous in machine learning, emerging as the standard approach in many settings. For example, they have been successfully applied for tasks such as topic prediction in citation networks [26]; molecule prediction [11]; and link prediction in recommender systems [33]. These applications typically make use of message-passing GNNs [11], whose idea is fairly simple: in each layer, nodes are updated by aggregating the information (messages) coming from their neighboring nodes.

Depending on how this aggregation is implemented, we can define different types of GNN layers. Two important and widely adopted layers are graph convolutional networks (GCNs) [18], which uniformly average the neighboring information; and graph attention networks (GATs) [30], which instead perform a weighted average, based on an attention score between receiver and sender nodes. More recently, a number of works have shown the strengths and limitations of both approaches from a theoretical [2, 3, 10], and empirical [19] point of view. These results show that their performance depends on the nature of the data at hand (i.e., the graph and the features), thus the standard approach is to select between GCNs and GATs via computationally demanding cross-validation.

In this work, we aim to exploit effectively and efficiently the benefits of both convolution and attention operations in the design of GNN architectures. To that end, we first introduce a novel graph convolutional attention layer (CAT), which extends existing attention layers by taking the convolved features as inputs of the score function, thus taking advantage of both operations. Following [10], we rely on a contextual stochastic block model to theoretically compare GCN, GAT, and CAT

architectures. Our analysis shows that, unfortunately, no free lunch exists among these three GNN architectures since their performance, as expected, is fully data-dependent.

This result motivates the main contribution of the paper, the *learnable graph convolutional attention network* (L-CAT): a novel GNN which, in each layer, is capable of automatically interpolating between the three operations during training by introducing only two additional (scalar) parameters. As a result, L-CAT is able to learn the proper operation to apply at each layer, thus combining different layer types in the same GNN architecture while overcoming the need to cross-validate—a process that was prohibitively expensive prior to this work. Our extensive empirical analysis demonstrates the capabilities of L-CAT on a wide range of datasets, outperforming existing baseline GNNs in terms of both performance, and robustness to input noise and network initialization.

## 2 Preliminaries

Assume we are given as an input an undirected graph $G = (V, E)$, where $V = [n]$ denotes the set of vertices of the graph, and $E \subseteq V \times V$ the set of edges. Each node $i \in [n]$ is represented by a $d$-dimensional feature vector $\mathbf{X}_i \in \mathbb{R}^d$, and the goal is to produce a set of predictions $\{\hat{\boldsymbol{y}}_i\}_{i=1}^n$.

To this end, a message-passing GNN layer yields, for each node $i$, a representation $\tilde{\boldsymbol{h}}_i \in \mathbb{R}^{d'}$, by collecting the information from each of its neighbors; aggregating them into a single message; and using the aggregated message to update its representation from the previous layer, $\boldsymbol{h}_i \in \mathbb{R}^d$. For the purposes of this work, we can define this operation as the following:

$$\tilde{\boldsymbol{h}}_i = f(\boldsymbol{h}_i') \quad \text{where} \quad \boldsymbol{h}_i' \stackrel{\text{def}}{=} \sum_{j \in N_i^*} \gamma_{ij} \boldsymbol{W}_v \boldsymbol{h}_j \ , \tag{1}$$

where $N_i^* = N_i \cup \{i\}$, and $N_i$ denotes the set of neighbors of node $i$, $\boldsymbol{W}_v \in \mathbb{R}^{d' \times d}$ a learnable weight matrix, $f$ an elementwise function, and $\gamma_{ij} \in [0, 1]$ are coefficients such that $\sum_{j \in N_i^*} \gamma_{ij} = 1$ for each node $i$.

Let the input features be $\boldsymbol{h}_i^0 = \mathbf{X}_i$, and the predictions be $\boldsymbol{h}_i^L = \hat{\boldsymbol{y}}_i$, we can readily define a message-passing GNN [11] as a sequence of $L$ layers as defined above. Depending on the way the coefficients $\gamma_{ij}$ are computed, we can identify different GNN flavors.

**Graph convolutional networks (GCNs)** [18] are a simple (yet effective) type of layers. In short, GCNs simply compute the average of the messages, i.e., they assign the same coefficient $\gamma_{ij} = 1/|N_i^*|$ to every neighbor:

$$\tilde{\boldsymbol{h}}_i = f(\boldsymbol{h}_i') \quad \text{where} \quad \boldsymbol{h}_i' \stackrel{\text{def}}{=} \frac{1}{|N_i^*|} \sum_{j \in N_i^*} \boldsymbol{W}_v \boldsymbol{h}_j \ , \tag{2}$$

**Graph attention networks** take a different approach. Instead of assigning a fixed value to each coefficient $\gamma_{ij}$, they dynamically compute it as a function of the sender and receiver nodes. A general formulation for these models can be written as follows:

$$\tilde{\boldsymbol{h}}_i = f(\boldsymbol{h}_i') \quad \text{where} \quad \boldsymbol{h}_i' \stackrel{\text{def}}{=} \sum_{j \in N_i^*} \gamma_{ij} \boldsymbol{W}_v \boldsymbol{h}_j \quad \text{and} \quad \gamma_{ij} \stackrel{\text{def}}{=} \frac{\exp(\Psi(\boldsymbol{h}_i, \boldsymbol{h}_j))}{\sum_{\ell \in N_i^*} \exp(\Psi(\boldsymbol{h}_i, \boldsymbol{h}_\ell))} \ . \tag{3}$$

Here, $\Psi(\boldsymbol{h}_i, \boldsymbol{h}_j) \stackrel{\text{def}}{=} \alpha(\boldsymbol{W}_q \boldsymbol{h}_i, \boldsymbol{W}_k \boldsymbol{h}_j)$ is known as the *score function* (or *attention architecture*), and measures the similarity between the messages $\boldsymbol{h}_i$ and $\boldsymbol{h}_j$ (or more generally, between a learnable mapping of the messages). From these scores, the (attention) coefficients are obtained by normalizing them, such that $\sum_j \gamma_{ij} = 1$. We can find in the literature different attention layers. Throughout this work, we focus on two types, the original GAT [30], and its extension GATv2 [5]:

$$\text{GAT:} \quad \Psi(\boldsymbol{h}_i, \boldsymbol{h}_j) = \text{LeakyRelu}\left(\boldsymbol{a}^\top [\boldsymbol{W}_q \boldsymbol{h}_i || \boldsymbol{W}_k \boldsymbol{h}_j]\right) \ , \tag{4}$$

$$\text{GATv2:} \quad \Psi(\boldsymbol{h}_i, \boldsymbol{h}_j) = \boldsymbol{a}^\top \text{LeakyRelu}\left(\boldsymbol{W}_q \boldsymbol{h}_i + \boldsymbol{W}_k \boldsymbol{h}_j\right) \ , \tag{5}$$

where the learnable parameters are now the attention vector $\boldsymbol{a}$; and the matrices $\boldsymbol{W}_q$, $\boldsymbol{W}_k$, and $\boldsymbol{W}_v$. Following previous work [5, 30], we assume that these matrices are coupled, i.e., $\boldsymbol{W}_q = \boldsymbol{W}_k = \boldsymbol{W}_v$.

Note that the difference between the two layers lies in the position of the vector $\boldsymbol{a}$: by taking it out of the nonlinearity, Brody et al. [5] increased the expressiveness of GATv2. Now, the product of $\boldsymbol{a}$ and a weight matrix does not collapse into another vector. More importantly, the addition of two different attention layers will help us show the versatility of the proposed models later in §6.

## 3  Previous work

In recent years, there has been a surge of research in GNNs. Here, we discuss other GNN models, attention mechanisms, and the recent findings on the limitations of GCNs and GATs.

The literature on GNNs is extensive [4, 14, 21, 34], and more abstract definitions of a message-passing GNN are possible, leading to other lines of work trying different ways to compute messages, aggregate them, or update the final message [7, 13, 35]. Alternatively, another line of work fully abandons message-passing, working instead with higher-order interactions [22]. While some of this work is orthogonal—or directly applicable—to the proposed model [7, 13, 35], here we focus on convolutional and attention graph layers, as they are the most widely used (and cited) as of today.

While we consider the original GAT [30] and GATv2 [5], our work can be directly applied to any attention model that sticks to the formulation in Eq. 3. For example, some works propose different metrics for the score function, like the dot-product [5], cosine similarity [28], or a combination of various functions [17]. Other works introduce transformer-based mechanisms [29] based on positional encoding [9, 20] or on the set transformer [31]. Finally, there also exist attention approaches designed for specific type of graphs, such as relational [6, 37] or heterogeneous graphs [16, 32].

### 3.1  On the limitations of GCN and GAT networks

In [2], the authors study classification on a Gaussian mixture, where the data correspond to the node features of a stochastic block model. They showed that when the graph is neither too sparse nor noisy, applying one layer of graph convolution increases the regime in which the data is linearly separable. Namely, if the distance between the means of the classes is not too small, the convolved features are linearly separable, whilst the original features are not. However, the above result is highly sensitive to the graph structure. Indeed, even if the distance between the means is large, the convolution cannot make the data linearly separable when the graph is noisy, since the convolution operation essentially collapses the means of the two classes to the same value.

More recently, Fountoulakis et al. [10] showed that GAT is able to remedy the above issue, and provide perfect node separability regardless of the noise level in the graph. Specifically, they showed that if the distance between the means is large compared to the standard deviation, then GAT achieves perfect node separability with high probability. However, a classical argument (see [1]) states that in this setting graph-based models are unnecessary, since a simple linear classifier already achieves perfect separability (see Proposition 4 in [10]). In addition, when the distance between the means is small compared to $\sigma$, no score function $\Psi$ can drop inter-class edges (the noisy edges), and thus might not achieve perfect node separability (see Conjecture 7 in [10]).

The above discussion implies that for some datasets, GAT might not work as well as expected. However, it leaves open the question of which architecture (GCN or GAT) is preferable in terms of performance.

## 4  Convolved attention: benefits and hurdles

In this section, we propose to combine attention with convolution operations. To motivate it, we complement the results of [10], providing a synthetic dataset for which *any* 1-layer GCN fails, but 1-layer GAT does not. Thus, proving a clear distinction between GAT and GCN layers. Besides, we show that convolution helps GAT as long as the graph noise is reasonable. The proofs for the two statements in this section appear in Appendix A and follow similar arguments as in [10].

This synthetic dataset is based on the *contextual stochastic block model* (CSBM) [8]. Let $\varepsilon_1, \ldots, \varepsilon_n$ be i.i.d uniform samples from $\{-1, 0, 1\}$. Let $C_k = \{j \in [n] \mid \varepsilon_j = k\}$ for $k \in \{-1, 0, 1\}$. We set the feature vector $\mathbf{X}_i \sim \mathcal{N}(\varepsilon_i \cdot \boldsymbol{\mu}, \mathbf{I} \cdot \sigma^2)$ where $\boldsymbol{\mu} \in \mathbb{R}^d$, $\sigma \in \mathbb{R}$, and $\mathbf{I} \in \{0, 1\}^{d \times d}$ is the identity matrix. For a given pair $p, q \in [0, 1]$ we consider the stochastic adjacency matrix $\mathbf{A} \in \{0, 1\}^{n \times n}$ defined as follows: for $i, j \in [n]$ in the same class (i.e., *intra-edge*), we set $a_{ij} \sim \text{Ber}(p)$;[1] for $i, j$ in different classes (i.e., *inter-edge*), we set $a_{ij} \sim \text{Ber}(q)$. We denote by $(\mathbf{X}, \mathbf{A}) \sim \text{CSBM}(n, p, q, \boldsymbol{\mu}, \sigma^2)$ a sample obtained according to the above random process. Our task is then to distinguish (or separate) nodes from $C_0$ vs. $C_{-1} \cup C_1$.

---

[1]$\text{Ber}(\cdot)$ denote the Bernoulli distribution.

Note that it is impossible to separate $C_0$ from $C_{-1} \cup C_1$ with a linear classifier (with high probability). In addition, by applying similar arguments as in [2], using one convolutional layer is detrimental for node classification on the CSBM.[2] This follows from the fact that although the convolution brings the means closer and shrinks the variance, the geometric structure of the problem does not change. On the other hand, we prove that GAT is able to achieve perfect node separability when the graph is not too sparse:

**Theorem 1.** Suppose that $p, q = \Omega(\log^2 n/n)$ and $\|\boldsymbol{\mu}\|_2 = \omega(\sigma\sqrt{\log n})$. Then, there exists a choice of attention architecture $\Psi$ such that, with probability at least $1 - o_n(1)$ over the data $(\mathbf{X}, \mathbf{A}) \sim \mathsf{CSBM}(n, p, q, \boldsymbol{\mu}, \sigma^2)$, GAT separates nodes $C_0$ from $C_1 \cup C_{-1}$.

Moreover, we show using methods from [2], that the above classification threshold $\|\boldsymbol{\mu}\|$ can be improved when the graph noise is reasonable. Specifically, *by applying convolution prior to the attention score*, the variance of the data is greatly reduced, and if the graph is not too noisy, the operation dramatically lowers the bound on $\|\boldsymbol{\mu}\|$ in Theorem 1. Motivated by this, we introduce the *graph convolutional attention layer* (CAT), which formalizes this idea:

$$\Psi(\boldsymbol{h}_i, \boldsymbol{h}_j) = \alpha(\boldsymbol{W}\tilde{\boldsymbol{h}}_i, \boldsymbol{W}\tilde{\boldsymbol{h}}_j) \quad \text{where} \quad \tilde{\boldsymbol{h}}_i = \frac{1}{|N_i^*|}\sum_{\ell \in N_i^*} \boldsymbol{h}_\ell \,, \tag{6}$$

and where $\tilde{\boldsymbol{h}}_i$ are the convolved features of the neighborhood of node $i$. As we show now, CAT improves over GAT by combining convolutions with attention, when the graph noise is low.

**Corollary 2.** Suppose $p, q = \Omega(\log^2 n/n)$ and $\|\boldsymbol{\mu}\| \geq \omega\left(\sigma\sqrt{\frac{(p+2q)\log n}{n(p-q)^2}}\right)$. Then, there is a choice of attention architecture $\Psi$ such that, with probability at least $1 - o(1)$ over the data $(\mathbf{X}, \mathbf{A}) \sim \mathsf{CSBM}(n, p, q, \boldsymbol{\mu}, \sigma^2)$, CAT separates nodes $C_0$ from $C_1 \cup C_{-1}$.

The above proposition shows that under the CSBM data model, convolving prior to attention changes the regime for perfect node separability by a factor of $|p - q|\sqrt{n/(p + 2q)}$. This is desirable when $|p - q|\sqrt{n/(p + 2q)} > 1$, since the regime for perfect classification is increased. Nonetheless, when $|p - q|\sqrt{n/(p + 2q)} < 1$, applying convolution prior to attention reduces the regime for perfect separability. Therefore, it is not always clear whether convolving prior to attention is beneficial.

## 5 L-CAT: Learning to interpolate

From the previous analysis, we can conclude that it is hard to know *a priori* whether attention, convolution, or convolved attention, will perform the best. In this section, we argue that this issue can be easily overcome by learning to interpolate between the three.

First, notice that the formulations of GCN and GAT only differ in that GCN weighs all neighbors equally (Eq. 2) and, the more similar the attention scores are (Eq. 3), the more uniform the coefficients $\gamma_{ij}$ will be. Thus, we can interpolate between GCN and GAT by introducing a learnable parameter $\lambda_1 \in [0, 1]$. Similarly, the formulation of GAT (Eq. 3) and CAT (Eq. 6) differ in the convolution within the score, which can be interpolated by another learnable parameter $\lambda_2 \in [0, 1]$.

Following this observation, we propose the *learnable convolutional attention layer* (L-CAT), which can be formulated as an attention layer with the following score:

$$\Psi(\boldsymbol{h}_i, \boldsymbol{h}_j) = \lambda_1 \cdot \alpha(\boldsymbol{W}\tilde{\boldsymbol{h}}_i, \boldsymbol{W}\tilde{\boldsymbol{h}}_j) \quad \text{where} \quad \tilde{\boldsymbol{h}}_i = \frac{\boldsymbol{h}_i + \lambda_2 \sum_{\ell \in N_i} \boldsymbol{h}_\ell}{1 + \lambda_2|N_i|} \,, \tag{7}$$

and where $\lambda_1, \lambda_2 \in [0, 1]$. As mentioned before, this formulation lets L-CAT learn to interpolate between GCN ($\lambda_1 = 0$), GAT ($\lambda_1 = 1$ and $\lambda_2 = 0$), and CAT ($\lambda_1 = 1$ and $\lambda_2 = 1$).

Despite its simplicity, L-CAT enables a number of non-trivial benefits. Not only can it switch between existing layers, but it can also learn to use the amount of attention necessary for each use-case. Moreover, by comprising the three layers in a single learnable formulation, it removes the necessity of cross-validating the type of layer, since their performance is data-dependent (see §§3.1 and 4). More importantly, it eases the task of combining different layer types within the same architecture.

---

[2]We note that this problem can be easily solved by two layers of GCN [3].

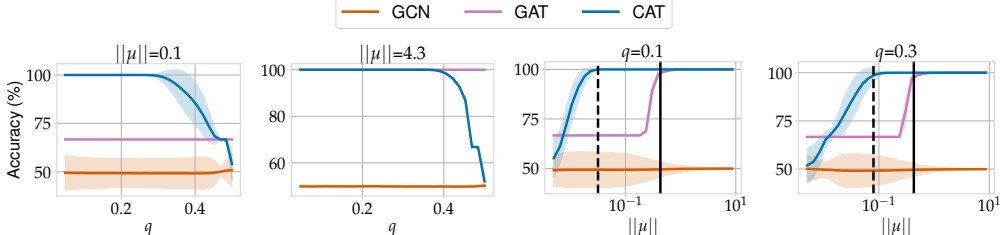

Figure 1: Synthetic data results. In the two left-most figures, we show how the accuracy varies with the noise level $q$ for $\|\boldsymbol{\mu}\| = 0.1$ and $\|\boldsymbol{\mu}\| = 4.3$. In the two right-most figures, we show how the accuracy varies with the norm of the means $\|\boldsymbol{\mu}\|$ for $q = 0.1$ and $q = 0.3$. We use two vertical lines to present the classification threshold stated in Theorem 1 (solid line) and Corollary 2 (dashed line).

## 6 Experiments

In this section, we assess the performance of the proposed models, CAT and L-CAT. First, we validate our theoretical findings on synthetic data (§6.1). Second, we compare all methods in various small-scale node classification tasks (§6.2). And finally, we evaluate the proposed models on more realistic scenarios from the Open Benchmark Graph [15] framework, assessing their performance and robustness to feature noise and network initialization (§6.3).

### 6.1 Synthetic data

In this section, we empirically validate our theoretical results (Theorem 1 and Corollary 2). We aim to better understand the behavior of each layer as the properties of the data change, i.e., the noise level $q$ (proportion of inter-edges) and the distance between the means of consecutive classes $\|\boldsymbol{\mu}\|$. We provide in Appendix B extra results and additional experiments.

**Experimental setup.** As data model, we use the proposed CSBM (see §4) with $n = 10000$, $p = 0.5$, $\sigma = 0.1$, and $d = n/\left(5 \log^2(n)\right)$. All results are averaged over $50$ runs, and parameters are set as described in Appendix A. To assess the sensitivity to structural noise, we perform two experiments. First, we vary the noise level $q$ between $0$ and $0.5$, leaving the mean vector $\boldsymbol{\mu}$ fixed. We test two values of $\|\boldsymbol{\mu}\|$: the first corresponds to the *easy* regime ($\|\boldsymbol{\mu}\| = 10\sigma\sqrt{2 \log n}$) where classes are far apart; and the second correspond to the *hard* regime ($\|\boldsymbol{\mu}\| = \sigma$) where the distance between the clusters is small. In the second experiment we modify instead the distance between the means, sweeping $\|\boldsymbol{\mu}\|$ in the range $\left[\sigma/20, 20\sigma\sqrt{2 \log n}\right]$, and thus covering the transition from the hard setting (small $\|\boldsymbol{\mu}\|$) to the easy one (large $\|\boldsymbol{\mu}\|$). Here, we fix $q$ to $0.1$ (low noise) and $0.3$ (high noise). In both cases, we compare the behavior of 1-layer GAT and CAT, and include GCN as the baseline.

**Results.** The two left-most plots of Fig. 1 show the node classification performance for the hard and easy regimes, respectively, as we vary the noise level $q$. In the hard regime, we observe that GAT is unable to achieve separation for any value of $q$, whereas CAT achieves perfect classification when $q$ is small enough. This exemplifies the advantage of CAT over GAT as stated in Corollary 2. When the distance between the means is large enough, we see that GAT achieves perfect results independently of $q$, as stated in Theorem 1. In contrast, as we increase $q$, CAT fails to satisfy the condition in Corollary 2, and therefore achieves inferior performance.

The results for the second set of experiments, where we fix $q$ and sweep $\|\boldsymbol{\mu}\|$, are shown in the right-most part of Fig. 1. In these two plots, we can appreciate the transition in the accuracy of both GAT and CAT as a function of $\|\boldsymbol{\mu}\|$. We observe that GAT achieves perfect accuracy when the distance between the means satisfies the condition in Theorem 1 (solid vertical line in Fig. 1). Moreover, we can see the improvement CAT obtains over GAT. Indeed, when $\|\boldsymbol{\mu}\|$ satisfies the conditions of Corollary 2 (dashed vertical line in Fig. 1), the classification threshold is improved. As we increase $q$ we see that the gap between the two vertical lines decreases, which means that the improvement decreases as $q$ increments, exactly as stated in Corollary 2.

These results—along with empirical evidence in the next sections—reinforce the idea that there is *a priori* no way to tell which layer to use, as their performance highly depend on the properties of the data. Prior to this work, this has been solved by cross-validating the layer type. In the next sections,

Table 1: Test accuracy (%) of the considered convolution and attention models for different datasets (sorted by their average node degree), and averaged over ten runs. Bold numbers are statistically different to their baseline model ($\alpha = 0.05$). Best average performance is underlined.

| Dataset | Amazon Computers | Amazon Photo | GitHub | Facebook PagePage | Coauthor Physics | TwitchEN |
|---|---|---|---|---|---|---|
| Avg. Deg. | 35.76 | 31.13 | 15.33 | 15.22 | 14.38 | 10.91 |
| GCN | 90.59 ± 0.36 | 95.13 ± 0.57 | 84.13 ± 0.44 | 94.76 ± 0.19 | 96.36 ± 0.10 | 57.83 ± 1.13 |
| GAT | 89.59 ± 0.61 | 94.02 ± 0.66 | 83.31 ± 0.18 | 94.16 ± 0.48 | 96.36 ± 0.10 | 57.59 ± 1.20 |
| CAT | **90.58 ± 0.40** | **94.77 ± 0.47** | **84.11 ± 0.66** | **94.71 ± 0.30** | 96.40 ± 0.10 | 58.09 ± 1.61 |
| L-CAT | **90.34 ± 0.47** | **94.93 ± 0.37** | **84.05 ± 0.70** | **94.81 ± 0.25** | 96.35 ± 0.10 | 57.88 ± 2.07 |
| GATv2 | 89.49 ± 0.53 | 93.47 ± 0.62 | 82.92 ± 0.45 | 93.44 ± 0.30 | 96.24 ± 0.19 | 57.70 ± 1.17 |
| CATv2 | **90.44 ± 0.46** | **94.81 ± 0.55** | **84.10 ± 0.88** | **94.27 ± 0.31** | 96.34 ± 0.12 | 57.99 ± 2.02 |
| L-CATv2 | **90.33 ± 0.44** | **94.79 ± 0.61** | **84.31 ± 0.59** | **94.44 ± 0.39** | 96.29 ± 0.13 | 57.89 ± 1.53 |

we empirically demonstrate that L-CAT can automatically perform layer selection during training, completely removing the need of cross-validating and, thus, saving computational resources.

## 6.2 Real data

In this section, we study the performance of the proposed models in a comprehensive set of real-world experiments, in order to gain further insights into the settings in which they excel. Specifically, we found CAT and L-CAT to outperform their baseline models as the average node degree increases. For a detailed description of the datasets and additional results, refer to Appendices C and D.

**Models.** We consider as baselines a simple GCN layer [18] where all neighbors are uniformly weighted, as well as the original GAT layer [30] and its recent extension, GATv2 [5]. See §2 for an introduction. Based on the two attention models, we consider their CAT-extensions, CAT and CATv2, as well as their interpolatable counterparts, L-CAT and L-CATv2. To ensure fair comparisons, all layers use the same number of parameters and share the same implementation, appropriately setting $\lambda_1$ and $\lambda_2$ (see Eq. 7) for each model.

**Datasets.** We take six node classification datasets. The *FacebookPagePage*/*GitHub*/*TwitchEN* datasets relate to social-network graphs [24], whose nodes represent verified pages/developers/streamers; and where the task is to predict the topic/expertise/explicit language usage of the node. The *Coauthor Physics* dataset [27] represents a co-authorship network whose nodes represent authors, and the task is to infer their main research field. Finally, the *Amazon Computers*/*Amazon Photo* datasets represent two product-similarity graphs [27], where each node is a product, and the task is to infer its category.

**Experimental setup.** To ensure the best results, we cross-validate all optimization-related hyperparameters for each model using GraphGym [36]. All models use four GNN layers with hidden size of 32, and thus have an equal number of parameters. For evaluation, we take the best-validation configuration during training, and report test-set performance. For further details, refer to Appendix D.

**Results** are presented in Table 1, averaged over 10 runs. In contrast with §6.1, we here find GCN to be a strong contender, reinforcing its viability in real-world data despite its simplicity. Moreover, we observe both CAT and L-CAT not only holding up the performance with respect to their baselines models for all datasets, but in most cases they also improve the test accuracy in a statistically significant manner. These results validate the effectiveness of CAT as a GNN layer, and show the viability of L-CAT as a drop-in replacement, achieving good results on all datasets.

As explained in §4, CAT differs from a usual GAT in that the score is computed with respect to the convolved features. Intuitively, this means that CAT should excel in those settings where nodes are better connected, allowing CAT to extract more information from their neighborhoods. Indeed, there exists a positive correlation between performance improvement and average degree of the graph. In the inset figure, we can observe the improvement in accuracy of CAT with respect to its baseline model, as a function of the average node degree of the datasets, and the linear regression of these results

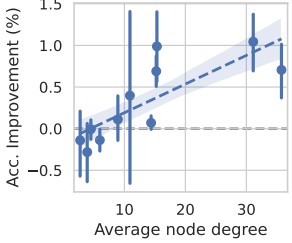

Table 2: Test performance of the considered convolutional and attention layers on four OGB datasets, averaged over five runs. Bold numbers are statistically different to their baseline model ($\alpha = 0.05$). Best average performance is underlined. Left table: accuracy (%); right table: AUC-ROC (%).

| Dataset | arxiv | products | mag | proteins |
|---------|-------|----------|-----|----------|
| GCN | $71.58 \pm 0.20$ | $74.12 \pm 1.20$ | $32.77 \pm 0.36$ | $80.10 \pm 0.55$ |
| GAT | $71.58 \pm 0.16$ | $78.53 \pm 0.91$ | $32.15 \pm 0.31$ | $79.08 \pm 1.47$ |
| CAT | $\mathbf{72.14 \pm 0.21}$ | $\mathbf{77.38 \pm 0.36}$ | $31.98 \pm 0.46$ | $73.26 \pm 1.65$ |
| L-CAT | $\mathbf{71.99 \pm 0.08}$ | $77.19 \pm 1.11$ | $32.47 \pm 0.38$ | $79.63 \pm 0.71$ |
| GATv2 | $71.73 \pm 0.24$ | $76.40 \pm 0.71$ | $32.76 \pm 0.18$ | $78.65 \pm 1.44$ |
| CATv2 | $\mathbf{72.03 \pm 0.09}$ | $74.81 \pm 1.12$ | $\mathbf{32.43 \pm 0.22}$ | $74.33 \pm 0.94$ |
| L-CATv2 | $71.97 \pm 0.22$ | $\mathbf{76.37 \pm 0.92}$ | $32.68 \pm 0.50$ | $\mathbf{79.07 \pm 0.98}$ |

247 (dashed line). This plot includes all datasets (from the manuscript and Appendix D), and shows a
248 positive trend between node connectivity and improved performance by CAT.

## 6.3 Open Graph Benchmark

250 In this section, we assess the robustness of the proposed models, in order to fully understand their
251 benefits. For further details and additional results, refer to Appendix E.

252 **Datasets.** We consider four different datasets from the Open Graph Benchmark (OGB) suite [15]:
253 *proteins*, *products*, *arxiv*, and *mag*. Note that these datasets are significantly larger than those from
254 §6.2 and correspond to more difficult tasks, e.g., *arxiv* is a 40-class classification problem (see Table 4
255 for details). This makes them more suitable for the proposed analysis.

256 **Experimental setup.** We adopt the same experimental setup as Brody et al. [5] for the *proteins*,
257 *products*, and *mag* datasets. For the *arxiv* dataset, we use instead the example code from OGB [15], as
258 it yields better performance than that of Brody et al. [5]. Just as in §6.2, we compare with GCN [18],
259 GAT [30], GATv2 [5], and their CAT and L-CAT counterparts. We cross-validate the number of
260 heads (1 and 8), repeat each experiment five times, and select the best-validation models during
261 training. All models share the network architecture, number of parameters, and network initialization.

262 **Results** are summarized in Table 2, averaged over 5 runs. Here we do not observe a clear preferred
263 baseline: GCN performs really well in *proteins* and *mag*; GAT excels in *products*; and GATv2 does
264 well in *arxiv* and *mag*. Let us now focus on the proposed models. While CAT obtains the best results
265 on *arxiv*, its performance on *proteins* and *products* is significantly worse than the baseline model.
266 Presumably, an excessive amount of inter-edges could explain why convolving the features prior to
267 computing the score is harmful, as seen in §6.1. As we explore in §6.3.2, however, CAT improves
268 over its baseline for most *proteins* scenarios, specially with a single head. In stark contrast, L-CAT
269 performs remarkably well, improving the baseline models in all datasets but *products*—even on those
270 in which CAT fails—demonstrating the adaptability of L-CAT to different scenarios.

271 In order to better understand the training dynamics of the different models, we plot in Fig. 2a the test
272 accuracy of GCN and the GATv2 models during training on the *arxiv* dataset. Interestingly, this plot
273 shows that while all models obtained similar final results, CATv2 and L-CATv2 drastically improved
274 their convergence speed and stability with respect to GATv2, matching that of GCN. To understand
275 the behavior of L-CATv2, we show in Fig. 2b the evolution of the parameters $\lambda_1, \lambda_2$. We observe
276 that to achieve these results, L-CATv2 converged to a GNN network that combines three types of
277 layers: the first layer is a CATv2 layer, taking advantage of the neighboring information; the second
278 layer is a quasi-GCN layer, in which scores are almost uniform and some neighboring information
279 is still used in the score; and the third layer is a pure GCN layer, in which all scores are uniformly
280 distributed. It is important to remark that these dynamics are fairly consistent, as L-CATv2 reached
281 the same $\lambda_1, \lambda_2$ values over all five runs.

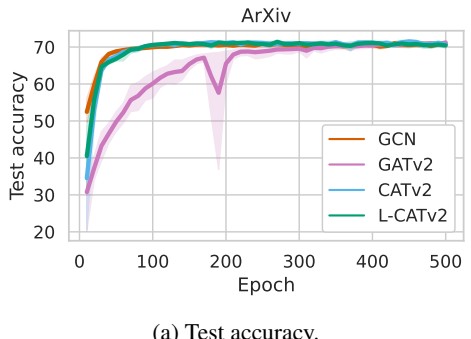
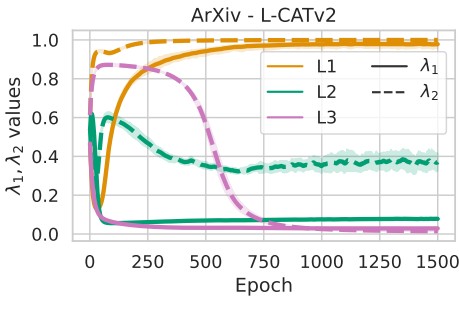

|         (a) Test accuracy.         |      (b) Evolution of $\lambda_1, \lambda_2$.      |

Figure 2: Behavior of GCN and GATv2 models during training on the *arxiv* dataset. *(a)* CAT and L-CAT converge quicker and are more stable than their baseline model. *(b)* L-CAT consistently converges to the same type of layers during training: a CAT →quasi-GCN→GCN network.

#### 6.3.1 Robustness to noise

One intrinsic aspect of real world data is the existence of noise. In this section, we explore the robustness of the proposed models to different levels of homoscedastic noise in the features. That is, we attempt to simulate scenarios where there exist measurement inaccuracies in the input features.

**Experimental setup.** For these experiments we consider the *arxiv* dataset, and the same experimental setup as in §6.3. To simulate homoscedastic noise, we introduce to the node features additive noise of the form $x' = x + \varepsilon$, where $\varepsilon \sim \mathcal{N}(\mathbf{0}, \mathbf{1}\sigma)$, and where we consider different levels of noise, specifically, we take $\sigma \in \{0, 0.25, 0.5, 0.75, 1\}$.

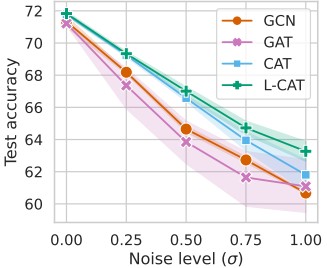

**Results** can be seen in the inset figure, which shows test accuracy as a function of the feature noise level $\sigma$. This plot summarizes the performance of all considered models, over five runs and two numbers of heads (1 and 8). We can observe that baseline attention models exhibit high variance and are quite sensitive to small perturbations. GCNs, instead, exhibit better robustness to noise and small variance. In concordance with the synthetic experiments (see §§4 and 6.1), we observe that CAT is able to leverage convolutions as a variance-reduction technique, boosting the performance of the attention mechanisms, and reducing their variance. Moreover, L-CAT proves to be strictly more robust than all other models: it boosts the performance and reduces the uncertainty—like CAT—and it is more effective than other approaches as it can adapt the amount of attention used in each layer, outperforming competing methods.

#### 6.3.2 Robustness to network initialization

Another important aspect of real-world applications is that of robustness to network initialization, i.e., the ability to obtain satisfying performance independently of the initial parameters. Otherwise, a practitioner could waste lots of resources trying different initilizations or, even worse, give up on a model just because they did not try the initial parameters that would yield great results. In this section, we test such a scenario using the *proteins* dataset as an example setting.

**Experimental setup.** We follow once again the same setup for *proteins* as in §6.3. We consider two different network initializations. The first one, *uniform*, uses uniform Glorot initilization [12] with a gain of 1, which is the standard initialization used throughout this work. The second one, *normal*, uses instead normal Glorot initialization [12] with a gain of $\sqrt{2}$. This is the initialization employed on the original GATv2 paper [5] exclusively for the *proteins* dataset.

**Results**—segregated by number of heads—are shown in Table 3, while the results for GCN appear in the inset table. These results show that the baseline models perform poorly on the *uniform* initialization. However, this is somewhat alleviated when using 8 heads in the attention models. Moreover, all baselines significantly improve with *normal* initialization,

|           |        GCN        |
|-----------|-------------------|
| *uniform* | $61.08 \pm 2.56$  |
| *normal*  | $80.10 \pm 0.55$  |
| average   | $70.59 \pm 10.21$ |

Table 3: Test AUC-ROC (%) on the *proteins* dataset for attention models with two different network initializations (see §6.3.2), using 1 head (top) and 8 heads (bottom).

|  |  | GAT | CAT | L-CAT | GATv2 | CATv2 | L-CATv2 |
|---|---|---|---|---|---|---|---|
| 1h | *uniform* | $59.73 \pm 3.61$ | $\mathbf{64.32 \pm 2.33}$ | $\mathbf{77.77 \pm 1.28}$ | $59.85 \pm 2.73$ | $\mathbf{64.32 \pm 2.33}$ | $\mathbf{79.08 \pm 0.95}$ |
|  | *normal* | $66.38 \pm 6.94$ | $73.26 \pm 1.65$ | $\mathbf{78.06 \pm 1.25}$ | $69.13 \pm 8.48$ | $74.33 \pm 0.94$ | $\underline{\mathbf{79.07 \pm 0.98}}$ |
| 8h | *uniform* | $72.23 \pm 2.86$ | $73.60 \pm 1.14$ | $\mathbf{78.85 \pm 1.57}$ | $75.21 \pm 1.61$ | $74.16 \pm 1.30$ | $\mathbf{78.77 \pm 0.97}$ |
|  | *normal* | $79.08 \pm 1.47$ | $\mathbf{74.67 \pm 1.15}$ | $\underline{79.63 \pm 0.71}$ | $78.65 \pm 1.44$ | $\mathbf{73.40 \pm 0.56}$ | $79.30 \pm 0.49$ |
|  | average | $69.36 \pm 8.52$ | $73.93 \pm 1.35$ | $78.58 \pm 1.48$ | $70.71 \pm 8.70$ | $71.55 \pm 4.54$ | $\underline{79.05 \pm 0.91}$ |

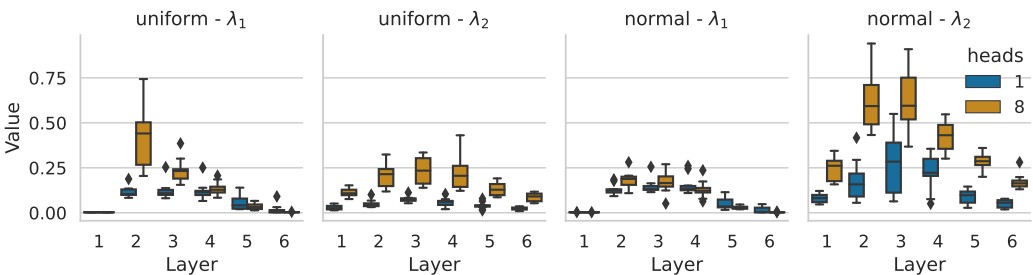

Figure 3: Distribution of $\lambda_1, \lambda_2$ on *proteins* dataset for L-CAT across initializations.

being GCN the best model, and attention models obtaining $79\%$ accuracy on average with $8$ heads. Compared to the baselines, CAT does a good job and improves the performance in all cases except for *normal* with $8$ heads. Remarkably, L-CAT consistently obtains a high accuracy in all scenarios and runs. This can be further appreciated by looking at the average accuracy (bottom row), showing that L-CAT is clearly more robust to parameter initialization than competing models.

To understand this performance, we inspect the distribution of $\lambda_1, \lambda_2$ for L-CAT in Fig. 3. Here, we can spot a few interesting patterns. First, the first and last layers are always GCNs, while the inner layers progressively admit less attention. Second, the number of heads affects the amount of attention allowed in the network; the more heads, the more expressive the layer tends to be, and the more attention that is permitted. Third, L-CAT adapts to the initialization used: in *uniform*, it stays competitive by allowing more attention in the second layer; in *normal*, it allows more attention in the score inputs. Table 3 and Fig. 3 thus consolidate the effectiveness and flexibility of L-CAT.

# 7 Conclusions and future work

In this work, we studied how to combine the strengths of convolution and attention layers in GNNs. We proposed CAT, which computes attention with respect to the convolved features, and analyzed its benefits and limitations on a new synthetic dataset. This analysis revealed different regimes where one model is preferred over the others, reinforcing the idea that selecting between GCNs, GATs, and now CATs, is a difficult task, as their performance directly depend on the data. For this reason, we proposed L-CAT, a model which is able to interpolate between the three via two learnable parameters. Extensive experimental results demonstrated the effectiveness of L-CAT, yielding great results while being more robust than other methods due to its adaptability. As a result, L-CAT proved to be a viable drop-in replacement that removes the need to cross-validate the layer type.

We do not consider this work adds any societal concerns. On the contrary, L-CAT eases the applicability of GNNs to the practitioner, and removes the need of cross-validating the layer type, which can potentially benefit other areas and applications, as GNNs have already proven.

We strongly believe learnable interpolation can get us a long way, and we hope L-CAT to motivate new and exciting work. For example, it would be interesting to see L-CAT applied to other GCN and GAT variants, such as those in [17, 28, 35]. Specially, we are eager to see L-CAT in real applications, and thus finding out what combining different GNN layers across a model (without the annoyance of cross-validating all combinations) can lead to in the real-world.

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
