# OpenReview forum: "Learnable Graph Convolutional Attention Networks"
_NeurIPS.cc/2022/Conference — NeurIPS 2022 Submitted_

### Official Review · Reviewer_QHuE · 2022-07-09

**Rating:** 6
**Confidence:** 5
**Soundness:** 4 excellent
**Presentation:** 4 excellent
**Contribution:** 2 fair

**Summary:**

This submissions has two parts of contribution.

First, it proposes a novel attention mechanism for graph neural networks. The proposed technique first aggregates neighborhood features (as done in GCN) before computing the attention weights. This new method is named CAT. By a theoretical, analysis, the authors show that CAT can be more effective than GAT but their comparison depends on the graph topology.

Second, for a given layer, the submission proposes to learn two additional scalar weights that interpolates between GCN , GAT, the newly proposed CAT. The intuition here is that the network can discover the correct network type to use depends on the input data.

Overall, I think the proposed method is novel and the theoretical analysis provides an interesting perspective. The empirical analysis shows that the proposed methods are more robust to data noise and weight initialization scheme. However, in the standard setting, the proposed method is mostly on-par with GCN (the comparison is not consistent across datasets). To take this empirical weakness into account, I am giving a weak accept.

**Questions:**

Questions:

1. As the authors acknowledged, many new variants of graph neural networks have been proposed. Can the authors include more baselines into comparison to contextualize the results better? I am especially interested in knowing how other baselines would react to the added feature noises.
2. Can the theretical analysis  also be applied to L-cat? Similarly, how would L-CAT behave on the synthetic data in Figure 1?
3. Another form of data noise is corruption to the graph adjacency matrix. Does L-CAT also handle this form of corruption better?

**Limitations:**

I do not see any negative societal impact.

**Strengths And Weaknesses:**

Strength

1. The proposed method is novel
2. The proposed method is motivated by a theoretical analysis. I appreciate the effort to understand the proposed method.
3. On the empirical side, the proposed method is more robust to data noise and different initialize schemes
4. The writing is very clear

Weakness

1. For the standard benchmarks, the proposed method is only on par with GCN, which is a simpler method.
2. I provide more requests and suggestions for improving the current draft in the questions section below.

---

> ### Author Response · Authors · 2022-07-31
> **Reply to reviewer QHuE**
>
> We want to thank the reviewer for the helpful suggestions on how to improve this work. We address the raised concerns below.
>
> **Theoretical analysis of L-CAT.** We omitted L-CAT from the theoretical analysis as it is not of interest from a theoretical point of view. As it is an interpolatable model between GCN, GAT, and CAT, for the theoretical analysis we can, in short, choose the ansatz that interpolates to the best performing-model for each particular case.
>
> **Extra experiments.** Let us remark that the focus of this work is not to outperform GCN (or other existing methods for that matter), but rather provide a simple-to-use alternative between convolutional and attention methods that is robust to different real-world scenarios. Our method is, moreover, applicable to other GCN models (see the [general reply](https://openreview.net/forum?id=2TdPjch_ogV&noteId=s5af2wEMY5C)). As part of these real-world scenarios, we agree that edge-noise is another scenario worth exploring. In fact, we have reproduced the results from Section 6.3.1 but corrupting the adjacency matrix (as in the [GATv2](https://arxiv.org/pdf/2105.14491.pdf) paper), and we observe that: i) CAT quickly drops in performance, matching that of GAT; and ii) L-CAT is consistently more robust than the rest of models. A new figure summarizing these results can be found [here](https://postimg.cc/4ntH00L7), and we will include these results in the next revision of the paper. Moreover, we observe interesting dynamics on the interpolations, reducing the amount of convolution as we introduce more noise (by modifying the lambda values). As of including more baselines, we will consider and welcome any particular baseline suggestion that the reviewer may have.

---

### Official Review · Reviewer_edc7 · 2022-07-10

**Rating:** 3
**Confidence:** 5
**Soundness:** 2 fair
**Presentation:** 3 good
**Contribution:** 2 fair

**Summary:**

The authors propose a new graph neural network by combining existing graph convolution networks: GCNs and GAT. The authors proposed a new learnable convolutional attention layer (L-CAT), which subsumes the graph convolution in GCNs, the graph attention layer in GAT, and a normalized attention layer (CAT).

**Questions:**

1. The experiments are somewhat obvious. The proposed method subsumes GCN, GAT, CAT. Then L-CAT must be on par with the models or outperform them.
2. In Table 1, L-CAT and CAT are comparable. The performance gain is marginal compared to GAT. Unless I missed something, both GCN and GAT are baselines of the proposed method.
3. In the literature, more powerful GNNs have been recently studied. Why do the authors compare their method with GCN?

**Limitations:**

1.  The proposed method is a simple generalized graph neural network that subsumes GCN and GAT layers. The simplicity of the proposed method is great as long as it has practical merits. However, the performance gain by the proposed method is marginal. In addition, the authors did not compare the proposed method with more recent works that exhibit stronger performance.
2. The proposed method is incremental, and the lack of generality of the proposed method makes the impact of this paper questionable.

**Strengths And Weaknesses:**

1. A new layer that subsumes a couple of popular graph convolution layers has been proposed. Without changing layers in GNNs, by optimizing parameters in the proposed layer, the proposed graph neural network can change its behavior.
2. This paper theoretically discussed the expressive power of GAT and the proposed layer CAT with a contextual stochastic block model (CSBM).

---

> ### Author Response · Authors · 2022-07-31
> **Reply to reviewer edc7**
>
> We would like to thank the reviewer for investing their time in reviewing our work. We discuss the raised concerns below.
>
> **Scope of the paper.** We would like to emphasize that the scope of this work is not to outperform existing methods, but instead to offer an effective and robust replacement that does not require expert fine-tuning. As such, our experiments show that: i) no method is preferred without prior knowledge of the data-generating process; ii) we can effectively learn to interpolate between different layers in a data-driven fashion, maintaining the performance, and allowing to combine different type of layers in the same GNN; and moreover iii) our method is more robust than existing approaches in terms of both feature noise (and edge noise, see the [reply to reviewer QHuE](https://openreview.net/forum?id=2TdPjch_ogV&noteId=UUe75PJ3gkr)), and network initialization. We strongly believe this work can be really impactful, simplifying the deployment pipeline of GNNs in real-world scenarios, and reducing the amount of cross-validation needed.
>
> **Generality.** In this work we focus on GCNs as they are broadly used, and GATs naturally extend them as shown in the main text. However, our method is applicable to other GCN approaches, and we will gladly add a new section explaining how to extend L-CAT to other architectures. Please refer to the [general response](https://openreview.net/forum?id=2TdPjch_ogV&noteId=s5af2wEMY5C) for further details.

---

> > ### Comment · Reviewer_edc7 · 2022-08-08
> > **The proposed method with other GCN approaches...**
> >
> > I thank the authors for the detailed responses. But my major concerns have not been addressed. Although the proposed method is applicable to other architectures in theory, no experimental results with other architectures were provided. Also, beyond the models (e.g., GCN and GAT) that are subsumed by the proposed model, it is not clear whether the proposed method will boost the performance of other GNN architecture or not. Further, even if the authors did not shoot for state-of-the-art performance, the baseline algorithms are definitely too weak. For these reasons, I maintain my original rating.

---

> > > ### Author Response · Authors · 2022-08-09
> > > **Reply to reviewer edc7**
> > >
> > > We thank the reviewer for the response and comments.  However, we still believe there is a misunderstanding regarding the contributions of our work.
> > >
> > > In relation to baselines, first, we would like to emphasize that previous work [1,2] has shown that the performance of GNN architectures highly depends on the dataset and the task to be solved. These results align with the theoretical findings not only in our work but in previous work [3]. Thus, to the best of our knowledge, there is no GNN model that outperforms others in all datasets and tasks. Second,  we would like to point out that we are comparing our model to GATv2 [4], a model that has just been accepted in ICLR 2022, which also uses as baselines GCN and GAT.  Third, we find that there is no general GNN architecture in the literature that is evaluated on the variety of datasets we used to evaluate our model (we use 15 different datasets). That said, if the reviewer could mention the concrete SOTA baselines that they refer to, we would be happy to include them in our experimental evaluation.
> > >
> > > Let us stress once again that the main contribution of our work is not to introduce a new GNN model that outperforms existing models, but instead, we provide a novel approach to interpolate between existing (and new) GNN models that allows automating model selection at each GNN layer of the architecture during learning. As shown in our results, our approach reduces the need for cross-validation without compromising performance and, more importantly, leads to results that are more robust to parameter initialization, and to noise introduced in the features and adjacency matrix.
> > >
> > >
> > > [1] Jiaxuan You, Zhitao Ying, and Jure Leskovec. Design space for graph neural networks. Advances
> > > in Neural Information Processing Systems (NeurIPS), 2020
> > >
> > > [2] Zhou, Kaixiong, et al. Auto-gnn: Neural architecture search of graph neural networks. arXiv preprint arXiv:1909.03184, 2019.
> > >
> > > [3] Kimon Fountoulakis, Amit Levi, Shenghao Yang, Aseem Baranwal, and Aukosh Jagannath. Graph attention retrospective. arXiv preprint arXiv:2202.13060, 2022.
> > >
> > > [4] S. Brody, U. Alon, and E. Yahav. How attentive are graph attention networks. In International Conference on Learning Representations (ICLR), 2022.

---

### Official Review · Reviewer_CqLh · 2022-07-11

**Rating:** 5
**Confidence:** 4
**Soundness:** 4 excellent
**Presentation:** 4 excellent
**Contribution:** 3 good

**Summary:**

In this paper, the authors first investigate the expressiveness of convolutional-based and attention-based GNN architectures under the CSBM setting, and propose graph convolutional attention (CAT) which hinges the attention with convolution. Knowing there is no clear winner between attention and convolution, the authors propose a technique to interpolate in the middle of all the three mechanisms, called L-CAT.

**Questions:**

- In “Graph Attention Retrospective”, the authors considered two-class CSBM. Is there any special reason that this paper considers three classes?

- CAT computes the attention score based on the neighborhood information of the two nodes. Does this mean CAT actually cheats using second-hop information? And naturally improve the expressiveness? Would it be more fair to compare CAT with two-layer GAT or GCN?


**Limitations:**

The analysis in this paper is fundamental and provides good insights. However, the major limitation lies in the performance on the benchmark datasets. According to the theoretical results in this paper, L-CAT generalizes GCN and GAT and learns to trade off between these two architectures. However, the final performance on OGB shows even GCN outperforms L-CAT on two datasets. Moreover, current SOTA models faraway outperform GCN and GAT. Trading off between GCN and GAT seems not a key to improve the performance.

**Strengths And Weaknesses:**

Strengths:

+ The paper is clearly written, well formulated, and easy to follow.

+ Both theoretical and empirical analysis on the separate power of attention using CSBM seems interesting, though largely followed from the “Graph Attention Retrospective” paper.

+ The ideas of combining and interpolating between convolution and attention are novel proposals and well-motivated.

+ The empirical evaluation justifies the theory on CSBM, demonstrates performance on both real data and benchmark, and presents many insightful visualization and ablation studies e.g., the learning curve and distributions of tunable parameters, robustness to initialization and noises.

Weaknesses:

- The theoretical analysis is based on toy settings. And the theoretical significance is not claimed very clearly given the “Graph Attention Retrospective” paper. Also the separation power of convolution-based architecture still remains obscure without a formal argument.

- The performance on the benchmark is not as good as expected. GCN and GAT seem to be a special case of L-CAT. However, L-CAT fails to outperform GCN on proteins and mag data. Also, lots of strong baselines are missing.

---

> ### Author Response · Authors · 2022-07-31
> **Reply to reviewer CqLh**
>
> We thank the reviewer for their review, kind words, and thought-provoking questions. We address all concerns below.
>
> **Similarities with “Graph Attention Retrospective”.** We are aware of the similarities with this work. In fact, we did not hide them. We acknowledge it (e.g., L36, L119), and it served as a driving-inspiration for _our own work_. As a consequence, the tools and proof techniques employed are similar to theirs. This is described in the main paper and repeated in the appendix (e.g., L426). We, however, employed them to analyse a different scenario, and the new proposed model (CAT).
>
> **3-class CSBM, GCNs, and theoretical significance.** We proposed the 3-class CSBM as a way of clearly understanding the expressivity power of GCN, GAT, and CAT. In appendix A.1 “A hard example for GCN”, we propose the new CSBM setting as it is not linearly separable (L436), and therefore a 1-layer GCN cannot separate the classes (L129, and L454). In this way, we can clearly show the benefits of combining convolutions—as a noise-reduction technique—with attention methods, in contrast with the 2-class CSBM previously considered, which is significantly important in our honest opinion. Moreover, GCNs were studied in prior work, and we extensively summarized these results in L95-102.
>
> **Fair comparisons with GCN.** We agree that CAT is naturally more expressive, and we tried to make comparisons as fair as possible. In the theoretical part, we restricted ourselves to 1-layer GNNs to better understand each layer type. While one may consider that CAT “cheats” by convolving beforehand, we showed that this additional step has both benefits and hurdles, depending on the inter-class noise level. On the experimental part, we cross-validate the number of layers of each model (while keeping the no. of parameters equal), ruling out any advantage that CAT could have on GCN in the 1-layer case.
>
> **Empirical results.** We would like to remind that the focus of our work is not obtaining state-of-the-art, but providing a drop-in replacement for broadly used methods, where the type of layer can be consistently learned from the data. As such, we show that the lambda parameters can be automatically learnt, matching the performance of existing models, while being more robust to different scenarios. Moreover, we would like to remark that GCN remain as a competitive method in existing baselines (e.g., SSP in [Cora](https://paperswithcode.com/sota/node-classification-on-cora) and [CiteSeer](https://paperswithcode.com/sota/node-classification-on-citeseer)). Regarding other baselines, we would gladly add any specific suggestion as time permits. Remarkably, our method is applicable to other baselines, even if we restricted ourselves to GCNs (see the [general response](https://openreview.net/forum?id=2TdPjch_ogV&noteId=s5af2wEMY5C)).

---

> > ### Comment · Reviewer_CqLh · 2022-08-09
> > **Thanks for the reply.**
> >
> > Thank for the authors' reply and sorry for the late response. All my questions have been well resolved. The most interesting point I found in this work is the theoretical extension to [1] under the "easy regime". Even though the result may look incremental, this paper successfully finds a scenario where the linear classifier no more works while GAT still has the separation power. This gives an insightful justification on when graph attention is necessary. It would be rather interesting to see the author extend their results to an arbitrary number of classes. Given all the above, I decide to raise my rating and advocate acceptance.
> >
> > [1] Fountoulakis et al. Graph Attention Retrospective, 2022

---

### Official Review · Reviewer_8bmS · 2022-07-12

**Rating:** 6
**Confidence:** 3
**Soundness:** 3 good
**Presentation:** 4 excellent
**Contribution:** 3 good

**Summary:**

The work developed by the authors in the paper consists in a design of a novel convolutional operation to achieve learning on graph data. This convolutional operation lies at the crossroad of "isotropic" GCN and "anisotropic"/ attentional GNN to take the advantages of both operations without increasing the complexity of the operation.
In this study, appropriate datasets and baselines are introduced to support the theoretical motivation and the robustness of the proposed operation.

**Questions:**

1/ Is the idea of combining GAT and GCN could be generalized to any pairs of convolutional operations on graphs ? or the proposed framework is restricted to these two convolutional operations

2/ Can you shed light on the fact that the proposed combination doesn't interfere with any recent convolutional operations design ?

3/ Could you compare (table 1, table 2) with recent convolutional operations ? it's a good starting that L-CAT outperforms GAT and GCN, however, one may wonder if it the case w.r.t to other convolutional operations.

**Limitations:**

The limitations are fairly discussed and the work does't have any societal concern.

**Strengths And Weaknesses:**

This work is a novel combination of two well known convolutional operations on unstructured data.  The combination is well motivated / studied methodologically and empirically. It can bring a value for a better understanding of learning on graphs and might give insight for the design of more expressive convolutional operations on graphs.
The authors studied theoretically the expressive powers of GCN and GAT while identifying their failures. Based on the latter, their combination is proposed in a straightforward manner while showing a gain in a performance without substantial increase of the complexity of GNN architectures. Moreover, their architecture is less sensitive to noisy data compared to GCN and GAT while being quasi-agnostic to the weights initialization which add significant value to the work. This direction could be interesting to explore further towards invariant architectures w.r.t weight initializations to make the networks more robust and get rid-off careful choice of initialization which is a tedious engineering task.

The related work is well studied and the experiments are compared with appropriate GNN models. Moreover, the paper is well structured and easy to follow, including the consistency of math variables.

I argue that the work is technically correct where the claims are well motivated theoretically and validated empirically with an appropriate experimental protocol. However, l am not sure to find some  mathematical parts developed  in the supplementary necessary to understand the contribution of the paper. Namely line 477 and 519.

One may wonder if the proposed framework can be generalized to any convolutional operations on graphs and to what extent it is possible or it is restricted to combining only GAT and GCN.  As a consequence, it's not clear if the proposed combination doesn't interfere with recent convolutional operations, including the seminal works on the design of convolutional operations from dynamical systems, partial differential equations and differential geometry standpoints.

---

> ### Author Response · Authors · 2022-07-31
> **Reply to reviewer 8bmS**
>
> We would like to thank the reviewer for the insightful and supportive review. Let us now address the raised concerns below.
>
> **Necessary mathematical derivations.** We are afraid not to follow the concerns regarding lines 477-519, and any further clarification on this point would be helpful for us. We find all the computations in the appendix correct and required for the analysis of the $\gamma$ values.
>
> **Generality of the proposed method.** Please refer to the [general answer](https://openreview.net/forum?id=2TdPjch_ogV&noteId=s5af2wEMY5C) for a detailed response on the generality of L-CAT. Regarding the interference with recent convolutional operations, as long as the models can be described in the general form described in the general answer (that is, their computations involve an adjacency matrix), L-CAT should be applicable. For example, L-CAT should work with [SE(3)-GNN](https://proceedings.mlr.press/v162/du22e/du22e.pdf) and [PDE-GCN](https://proceedings.neurips.cc/paper/2021/file/1f9f9d8ff75205aa73ec83e543d8b571-Paper.pdf), although this would require further investigation.
>
> **Add baselines to tables 1 and 2.** We would like to stress that the focus of our work is not to outperform existing methods, but to provide a robust drop-in replacement for, in this case, GCNs and GATs. As such, we believe our experiments are appropriate and sufficient, as acknowledged in the review. Moreover, as discussed in the general reply, L-CAT can be directly applied to other convolution operations and, thus, we fail to see how these results wouldn’t hold. If time permits, and a specific instance of recent convolutional operation is provided, we will work on adding results such as those in Tables 1 and 2 using a different base model.

---

### Author Response · Authors · 2022-07-31
**General reply**

We thank all the reviewing team for their invaluable feedback, which will definitely help improve the state of the paper. We are particularly happy to read that the work is _"well-motivated and easy to follow"_ (reviewers 8bmS, CqLh, QHuE), that _"the claims are validated empirically with an appropriate experimental protocol"_ (reviewer 8bmS), and that _"the analysis in this paper is fundamental and provides good insights"_ (reviewer CqLh). Below, we provide some general comments that we consider worth-clarifying.

**Impact and scope.** We would like to emphasize that the goal of our work is not to pursue state-of-the-art or outperform other GNN approaches. Instead, our work focuses on providing a drop-in replacement of convolutional and attention-based GNNs that reduces the need of cross-validating (via data-driven interpolation), while being as competitive as existing approaches and more robust to: network initialization; feature noise; and edge noise (this last result is new, as suggested by reviewer QHuE, follow [this link](https://postimg.cc/4ntH00L7) for a plot summarizing the new results). As such, we consider our work to have a great potential impact for practitioners. We will certainly work on emphasizing this message in the next revision.

**Generability of L-CAT.** In this work we focus on GCN and GAT for two main reasons: i) simplicity, as their interpolation is natural and easy-to-understand; and ii) potential impact, as they are broadly-used and heavily-extended (see, e.g., [connected papers](https://www.connectedpapers.com/main/36eff562f65125511b5dfab68ce7f7a943c27478/Semi%20Supervised-Classification-with-Graph-Convolutional-Networks/graph)). However, L-CAT can be easily extended to other GCNs and GAT extensions as follows:
- Convolutional operations. In essence, what we provide is an interpolation between an adjacency matrix $A$ and an attention-generated matrix $A^{\text{att}}$, where each entry of this attention matrix is given by $n \cdot \gamma_{ij}$ as in Eq. (3). Let us denote this interpolation matrix as $A_{\lambda_1, \lambda_2}^{\text{att}}$. Then, we can replace GCN with any other method by replacing the adjacency matrix in its formulation. For example, a GCN with the normalized adjacency matrix computes $Z = W\tilde{D}^{1/2}\tilde{A}\tilde{D}^{1/2}X$ where $\tilde{A} = I + A$ and $\tilde{D}$ is a diagonal matrix with $\tilde{D_{ii}} = \sum_j \tilde{A_{ij}}$. To apply L-CAT, we just need to replace $A$ by $A_{\lambda_1, \lambda_2}^{\text{att}}$. Note that this specific instance of using attention with other convolutional operations has been already successfully applied in the literature (e.g., see Eq. (10) from [Bag of Tricks for Node Classification with Graph Neural Networks](https://arxiv.org/abs/2103.13355v4)).
- Attention operations. In the main text we already show how to use recent extensions of GAT, namely, GATv2 (ICLR 2022), as generally it simply consists on changing the way attention coefficients are computed, which is orthogonal to our method.

We will add a new section in the paper explaining how to extend L-CAT to other architectures in full-detail.

---

### Meta-Review · Area_Chair_uQgS · 2022-08-26

**Recommendation:** Reject
**Confidence:** Less certain

**Metareview:**

The paper explores the advantages of both GCN and GAT by proposing a learnable network that can interpolate between GCN, GAT and CAT for each layer automatically. The proposed research idea is novel and discovers an interesting perspective by combining and interpolating between convolution and attention networks. The paper is theoretically sound, and extensive experiments are conducted over 15 datasets with a comprehensive analysis.

However, all the reviewers consistently raise concerns regarding incremental improvement compared with baselines, and another common concern is that the authors do not extend the proposed method with more advanced convolutional and attention networks. The authors argue that their intuition is to design a more robust replacement to GCN/GAT, not a SOTA. However, the author should be aware that L-CAT is able to extend to other networks does not guarantee it will work, and it is possible that the convolution and attention network may conflict during the training. Since the proposed method is a novel and general paradigm, solid experiments are needed to thoroughly evaluate its performance. The motivation is promising, but more experiments should be conducted to sufficiently prove the superiority of the proposed method.

**Award:**

No

---

### Decision · Program_Chairs · 2022-09-14

Reject